# Antimicrobial susceptibility of clinical *Helicobacter pylori* isolates and its eradication by standard triple therapy: a study in west central region of Colombia

Adalucy Alvarez-Aldana,[1] Paula Andrea Fernandez Uribe,[2] Tatiana Mejía Valencia,[3] Yina Marcela Guaca-Gonzalez,[4] Jorge Javier Santacruz-Ibarra,[4] Brenda Lucia Arturo-Arias,[5,6] Luis Javier Castañeda-Chavez,[7] Robinson Pacheco-López,[2] Lina María Londoño-Giraldo,[1] José Ignacio Moncayo-Ortiz[4]

**ABSTRACT**  The aim of the present study was first to isolate *Helicobacter pylori* from gastric biopsy specimens and to test their antibiotic susceptibility. Second, it was to evaluate the efficacy of the standard triple therapy from patients of the west central region of Colombia. *H. pylori* positive patients received standard triple therapy with proton pump inhibitor (PPI) (40 mg b.i.d.), clarithromycin (500 mg b.i.d.), and amoxicillin (1 g b.i.d.) for 14 days. Thereafter, antibiotic susceptibility of the isolates was assessed by E-Test. From 94 patients enrolled, 67 were positive for *H. pylori* by histology or culture. Overall resistance to metronidazole, levofloxacin, rifampicin, clarithromycin, and amoxicillin was 81%, 26.2%, 23.9%, 19%, and 9.5%, respectively. No resistance was found for tetracycline. A total of 54 patients received standard triple therapy, 48 attended follow-ups testing, and of them, 30 had resistance test reports. Overall eradication rate was 81.2%. Second-line treatment was given to eight patients, four of whom were followed up with a 13C urea breath test (UBT) and remained positive for *H. pylori*. Eradication was significantly higher in patients with clarithromycin susceptible than in resistant strains (95.6% vs 42.8% $P = 0.001$). The updated percentages of resistance to clarithromycin in this geographical area had increased, so this value must be considered when choosing the treatment regimen.

**IMPORTANCE**  Antibiotic resistance in *Helicobacter pylori* has increased worldwide, as has resistance to multiple antimicrobials (MDRs), which seriously hampers the successful eradication of the infection. The ideal success rate in eradicating *H. pylori* infection (≥90%) was not achieved in this study (81.2%). This is the first time that MDR is reported (14.3%) in the region; the resistance to clarithromycin increased over time (3.8%–19%), and levofloxacin (26.2%) and rifampicin (23%) resistant isolates were detected for the first time. With these results, strain susceptibility testing is increasingly important, and the selection of treatment regimen should be based on local antibiotic resistance patterns.

**KEYWORDS**  *Helicobacter pylori*, antibiotics, resistance, drug therapy, eradication, Colombia

*H*elicobacter pylori infection has a worldwide overall prevalence of 44%. This rate ranged from 50.8% in developing countries compared with 34.7% in developed countries (1). *H. pylori* is a microaerophilic Gram-negative bacterium that can survive in highly acidic environments and is generally acquired in the first 5 years of life by oral-oral or fecal-oral transmission, linked to of low socioeconomic status, poor hygiene, and overcrowding (2, 3).

Address correspondence to Adalucy Alvarez-Aldana, adalucy.alvareza@unilibre.edu.co.

The authors declare no conflict of interest.

See the funding table on p. 10.

*[This article was published on 25 June 2024 with an error in the affiliations. The affiliations were corrected in the current version, posted on 28 June 2024.]*

*H. pylori* causes persistent inflammation in the gastric mucosa (4) that increases the risk of clinical symptoms associated with peptic ulcers or chronic gastritis that may later progress to chronic forms leading to intestinal metaplasia, dysplasia, and ultimately, gastric adenocarcinoma (5, 6). *H. pylori* infection is consistently recognized as the most important risk factor for gastric cancer (7). Since 1994, *H. pylori* has been identified as a class-1 carcinogen by the International Agency for Research on Cancer (IARC) (8).

*H. pylori* eradication may rapidly decrease active inflammation in the gastric mucosa, prevent progression toward precancerous lesions and reverse gastric atrophy before the development of intestinal metaplasia. Undoubtedly, the earliest possible eradication of *H. pylori* is highly beneficial (9). *H. pylori* gastritis as an infectious disease is now included as a nosological entity in the new International Classification of Disease 11th Revision (ICD 11), which implies treatment of all *H. pylori*-infected patients. Eradication of *H. pylori* is recommended even in the absence of symptoms. Empirical *H. pylori* eradication included triple standard therapy. It consists of a proton pump inhibitor (PPI) in standard doses accompanied by two antibiotics such as clarithromycin plus amoxicillin for 14 days (10). Resistance of *H. pylori* to antibiotics has reached alarming levels worldwide, which has been identified as one of the main causes of therapeutic failure (11, 12). Amoxicillin, metronidazole, clarithromycin, tetracycline, and levofloxacin are the most frequent antibiotics used in different combinations in eradication regimens (13).

The current study first aimed to isolate *H. pylori* in cultures from gastric biopsy samples from patients coming of the western central region of Colombia and test their susceptibility to amoxicillin, clarithromycin, metronidazole, tetracycline, rifampicin, and levofloxacin. We also aimed to assess the efficacy of the standard triple therapy in the population from two specialized Health Centers located in the cities Pereira and Manizales. In this part of the country, like different regions, positive infected patients are treated in an empirical way, without previous analysis of antimicrobial drug resistance, which could increase the risk of emergence of multidrug resistant strains.

## MATERIALS AND METHODS

### Patient population

Consecutive adult patients with gastroduodenal disease who underwent gastroduodenoscopy and biopsy sampling at two specialized centers in the west-central region of Colombia (cities of Pereira and Manizales) between February and October 2018 and gave written informed consent to participate in the study were included. Patients with comorbidities, immunosuppression due to risk of infection, previous gastric surgery, and those who had used PPIs, anti-H2 inhibitors, or antibiotics 4 weeks before the study were excluded. Histological and culture studies were performed to diagnose *H. pylori*.

The study was approved by the Bioethics Committee (BC) of the Universidad Tecnológica de Pereira (Pereira-Colombia), and the BC approved the informed consent before the start of the project.

### *H. pylori* culture

*H. pylori* was cultured from gastric biopsy samples of the antrum, body, and fundus. Samples were stored in Brain Hearth Infusion (BHI) broth with 20% glycerol, supplemented with antibiotics (vancomycin 10 mg/L, polymyxin B 0.33 mg/L, bacitracin 1.07 mg/L, and amphotericin B 5 mg/L).

The biopsy specimens were mixed with sterile saline solution and macerated with a homogenizer (Deltaware Pellet Pestle). We kept the maceration tissues at −80°C in BHI broth with glycerol. Then 100 µL of each mash solution was plated onto culture media Tryptic soy agar (TSA) (Oxoid or Merck), supplemented with sheep blood (7%), isovitalex (0.5%), and the same antibiotics/concentrations used for the BHI broth media during transportation and were incubated under microaerophilic conditions (5% $O_2$, 10% $CO_2$ and 85% $N_2$) at 37°C for 5–7 days. Colonies were confirmed with Gram staining and biochemical tests (positive urease, catalase and oxidase test).

## E-test and definition of susceptibility testing

*H. pylori* isolates obtained by the primary culture were subcultured on non-selective TSA (Oxoid or Merck) with 7% sheep blood, 0.5% isovitalex (BBL). The isolates were subcultured to discover the profile of antimicrobial resistance by E-test (AB BIODISK North American Inc., Piscataway, NJ, USA) for metronidazole, clarithromycin, amoxicillin, levofloxacin, rifampicin, and tetracycline. Suspensions from pure 48 h subcultures were prepared in *Brucella* broth supplemented with 0.5% Isovitalex, and inoculum turbidity was adjusted to McFarland 3.0 standard. Thereafter, they were inoculated onto TSA plates supplemented with sheep blood (7%), isovitalex (0.5%), and without antibiotics. E-test strips were placed and incubated under microaerophilic conditions at 37°C for 72 h.

Antimicrobial activity was detected as a minimum inhibitory concentration (MICs). *H. pylori* strain ATCC 43504™ was used as a control. Clarithromycin MICs were interpreted based on CLSI breakpoints (≥1.0 mg/L resistant) (CLSI, 2016) (14); we also used EUCAST breakpoints (15) for amoxicillin ≥ 0.125 mg/L, levofloxacin ≥ 1 mg/L, tetracycline ≥ 1 mg/L, rifampin ≥ 1 mg/L, and metronidazole ≥ 8 mg/L antibiotics.

## Treatment and follow-up by $^{13}$C-urea breath test

*H. pylori* positive patients received triple therapy with PPI (omeprazole 40 mg b.i.d.), clarithromycin (500 mg b.i.d.), and amoxicillin (1 g b.i.d.) for 14 days. Forty-five days after the end of therapy, $^{13}$C urea breath test (13C UBT) (TAU-KIT, Isomed S.L., Madrid, Spain) was performed with citric acid and 100 mg of 13C-urea. Patients who were positive after completion of the first treatment were re-treated with the second-line regimen with PPI (40 mg b.i.d.), amoxicillin (1 g b.i.d.), tetracycline (500 mg t.i.d.), and bismuth subsalicylate (524 mg b.i.d.) for 14 and 45 days after the end of treatment, a $^{13}$C UBT was performed as a follow-up.

## Statistical analysis

For the analysis of eradication, the population included all *H. pylori* positive patients who received full doses of treatment and had 13C UBT for follow-up, while for the reporting of antibiotic susceptibility testing (AST), the population included *H. pylori* positive patients who did or did not receive pharmacological treatment. Data that did not meet the conditions were not considered. Correlational statistics were performed in the IBM SPSS Statistics v.25 program. Eradication rates between groups were compared by Chi 2 test or Fisher's exact test, and odds ratios (ORs) and 95% confidence intervals (CIs) were estimated. *P* values < 0.05 were considered significant.

## RESULTS

### Patient population

The sociodemographic variables of this population were reported in our previous publication (2). Between February and October 2018, 740 gastroenterology consultations were registered in two centers in Pereira and Manizales, of which 94 patients met the selection criteria. Six hundred and forty six individuals did not meet the selection criteria (580 patients met the exclusion criteria, 38 patients did not sign the informed consent or did not accept the procedure, and 28 did not present authorization from the health entity for the procedure) (Fig. 1 blue dashed line). Of all patients, none were younger than 18 years, 40 were aged 18–45 years (42.6%), 52 were 46–65 years old (55.3%), and 2 were older than 65 years (2.1%).

Female participants were 76.5%, and the mean age was 46 years (SD 10.4). Male participants were 24.5%, and the mean age was 42 years (SD 7.7). Most patients were housewives (48.9%), and 59.5% of them were from Pereira.

Regarding diagnostic tests, 71.2% (67/94) of the patients had at least one positive test (histology or culture). They were positive for both tests 55.2% (37/67), positive by culture

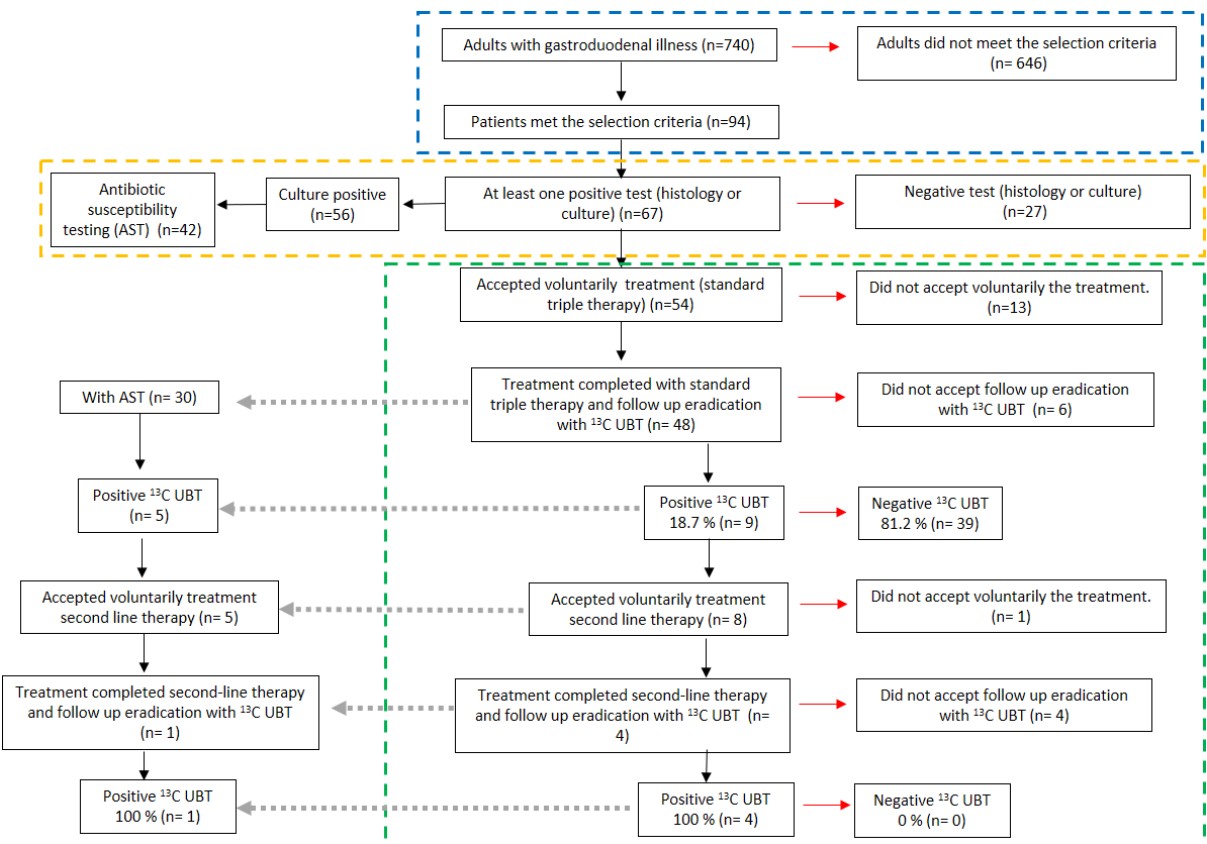

**FIG 1** General scheme of patients analyzed with *H. pylori* infection and eradication treatment. Blue dashed line enclose patients met selection criteria, orange dashed line enclose positive *H. pylori* patients, green dashed line enclose *H. pylori* eradication rate of patients and gray dotted arrows point to patients with completed treatment, breath testing and antibiotic susceptibility profiling.

28.4% (19/67), and positive by histological examination 16.4% (11/67). Culture detected 83.6% (56/67), and histological examination detected 71.6% (48/67) of *H. pylori* positive patients. Of the positive patients, 77.6% (52/67) were women and 22.3% (15/67) were men. An AST report was obtained in 75% (42/56) of the patients with positive cultures (Fig. 1 orange dashed line). The 14 primary culture isolates without AST data were due to lack of growth or contamination. Histologically, 25.5% (24/94) patients had malignancy precursors, being 37.5% (9/24) *H. pylori* positive while in the other 74.5% (70/94) patients without malignancy precursors, 61.4% (43/70) were positive. Of all patients, gastropathy, chronic antral or corporal gastritis, duodenal or gastric ulcer, precursor lesions of malignancy, and atrophic gastritis of the antrum were found in 71.6% (48/67), 13.4% (9/67), 7.5% (5/67), 6% (4/67), and 1.5% (1/67), respectively.

Of all the *H. pylori* positive patients (*n* = 67), 54 voluntarily received triple therapy; of these, 48 patients voluntarily agreed to be followed up with 13C UBT, constituting the group for the analysis of *H. pylori* eradication. Six of 54 (11.1%) patients reported side effects of therapy (nausea), with no interference with daily activities.

### *H. pylori* eradication rate of patients

Of the 48 patients, with a positive report for *H. pylori* who received antibiotic management and underwent follow-up with a breath test after finishing the pharmacological management, 9 patients had a positive result, providing an eradication percentage with standard triple therapy of 81.2% (39/48). Eight of the nine positive patients accepted pharmacological medical treatment with second-line therapy; of these, four patients accepted follow-up with breath tests that remained positive (Fig. 1 green dashed line).

## Antimicrobial susceptibility

Of the 56 *H. pylori* positive patients by culture, antibiotic susceptibility profiles were obtained for 42 patients (75%) (14 primary culture isolates without AST data were due to the lack of growth or contaminating bacteria that can inhibit the growth of *H. pylori*, especially if is of low density in the sample). All isolates were susceptible to tetracycline, metronidazole, levofloxacin, rifampicin, clarithromycin, and amoxicillin, and dual, triple, quadruple resistance was found (Table 1).

Table 2 gives an overview of overall resistance rates, the number of resistant isolates, and distribution of the MICs (mg/L) found in the 42 isolates. Briefly, according to the minimal inhibitory concentrations, 66% (23/35) and 38% (3/8) of the isolates showed MIC values over 256 mg/L for metronidazole and clarithromycin, respectively.

When comparing the results previously obtained by this research team in the Central Western Region of Colombia (data published in 2009 and 2020) (16, 17) and the results described in this study, a stable percentage of the rate of resistance to metronidazole was observed. In the case of clarithromycin, there was an increase of more than double, finding a statistically significant difference between 2009 and the present study (*P*-value = 0.004). No resistance to amoxicillin was found in 2020 study (16), while in 2009 study (17) and in the current study were 1.9% and 9.5%, respectively. There is a statistically significant difference between 2020 and the current study (P-value = 0.02). In this report, no tetracycline resistance was found, which is consistent with what was found in previous studies. There are no earlier data in the region on the presence of resistance related to levofloxacin and rifampicin, so it was not possible to make a comparison. In the rest of the comparisons, no significant statistical difference was found. Resistance to two or more antibiotics had a significant increase for the antibiotics evaluated in the previous studies (Table 3).

## Efficacy of triple therapy and effects of drug resistance

Of the 67 *H. pylori* positive patients, 54 received standard triple therapy, and 48 patients underwent follow-up breath testing. Of these 48 patients (both with completed treatment and breath testing), antibiotic susceptibility profiling was obtained in 30 patients. Missing data refer to samples with lack of growth or contaminating bacteria that can inhibit the growth of *H. pylori*, especially if it is of low density in the sample. Table 4 describes the eradication rates according to the resistance profile of these 30 patients.

**TABLE 1**  Results of the susceptibility test of 42 *H. pylori* isolates

| Antibiotics | Isolates | Rate (%) |
|---|---|---|
| Metronidazole | 15 | 35.7 |
| Rifampicin | 3 | 7.1 |
| Levofloxacin | 2 | 4,8 |
| Clarithromycin | 0 | 0 |
| Amoxicillin | 0 | 0 |
| Metronidazole + Levofloxacin | 3 | 7.1 |
| Metronidazole + Clarithromycin | 4 | 9.5 |
| Metronidazole + Rifampicin | 4 | 9.5 |
| Metronidazole + Amoxicillin | 2 | 4.8 |
| Levofloxacin + Clarithromycin | 1 | 2.4 |
| Metronidazole + Levofloxacin + Rifampicin | 3 | 7.1 |
| Metronidazole + Clarithromycin + Amoxicillin | 1 | 2.4 |
| Metronidazole + Clarithromycin + Levofloxacin | 1 | 2.4 |
| Metronidazole + Clarithromycin + Levofloxacin + Amoxicillin | 1 | 2.4 |
| Not resistant | 2 | 4.8 |
| TOTAL | 42 | 100.0 |

**TABLE 2** Distribution of overall resistance rates and MICs in *H. pylori* isolates by E-test method[a]

| E-test | Number of isolates | | | | | |
|---|---|---|---|---|---|---|
| MIC, mg/L | MTZ | CLA | AMX | LEV | RIF | TET |
| <0.125 | 3 | 33 | 38 | 18 | 7 | 36 |
| ≥0.125–0.99 | 2 | 1 | 4 | 13 | 25 | 6 |
| ≥1.0–7.9 | 2 | 1 | 0 | 0 | 9 | 0 |
| ≥8.0–31.9 | 8 | 0 | 0 | 0 | 1 | 0 |
| ≥32–128 | 4 | 4 | 0 | 11 | 0 | 0 |
| ≥ 256 | 23 | 3 | 0 | 0 | 0 | 0 |
| Resistance, % (*n*) | 83.3 (35/42) | 19 (8/42) | 9.5 (4/42) | 26.2 (11/42) | 23.8 (10/42) | 0 (0/42) |
| 95% CI | 72.1–94.6 | 7.2–31.0 | 6.5–18.4 | 12.9–39.5 | 11.0–36.7 | |

[a]MIC, minimum inhibitory concentration; MTZ, metronidazole; CLA, clarithromycin; AMX, amoxicillin; LEV, levofloxacin; RIF, rifampicin; TET, tetracycline.

Of these 30 patients who had the complete triple therapy, follow-up by breath test and antibiotic susceptibility profile, eradication failed in five patients (16%). The susceptibility profile of the strains from these five patients was as follows: resistance to metronidazole/clarithromycin (two patients), resistance to clarithromycin/levofloxacin (one patient), and resistance to three or more antibiotics (two patients). Of these five patients, five accepted second-line treatment, and one had a 13C UBT follow-up and remained positive due to metronidazole/clarithromycin/amoxicillin resistant strain (Fig. 1 gray dotted arrows).

Among the remaining 12 susceptibility profiles belonging to patients who refused treatment or did not accept follow-up with 13C UBT, 10 had resistance to one or more antibiotics (Table 1).

No significant differences were found in the mean eradication rates between patients with metronidazole resistant and susceptible strains (84% and 80% respectively); however, eradication was significantly higher in patients with clarithromycin susceptible than in resistant strains (95.6% vs 42.8% $P = 0.001$). Finally, of the five strains that did not respond to triple standard therapy, four had combined resistances to clarithromycin and metronidazole with MICs of 1.5, 64, 256, 64 mg/L and 256, 12, 16, 24 mg/L, respectively.

## DISCUSSION

In this study, triple standard therapy for 14 days was effective for the eradication of *H. pylori* in 81.2% of the patients with a positive histology or culture, which is similar to the results in other studies in Colombia that reported the eradication success in about 80% of the cases (18). The goal of eradication therapy is to reliably cure *H. pylori* infection in the majority (≥90%) of patients (10).

It is important to highlight that, of the positives, 83.6% of the patients were detected by culture and 71.6% of them by histological study, finding that the culture detection rate was the same that was reported by others (50%–95%) (19). There were 16.4% false negative cultures, which could be associated with incubation <10 days, some studies recommend prolonged incubation for some strains, especially those enduring hostile environment or a period of antibiotic force (20, 21). *H. pylori* could transform from the

**TABLE 3** Comparison of previous and current antimicrobial resistance in *H. pylori* isolates

| Antibiotic | | | Resistance, *n* (%) | | |
|---|---|---|---|---|---|
| | 2009 | 2020 | Current | *P*-value 2009 vs current | *P*-value 2020 vs current |
| | (*n* = 106) | (*n* = 61) | (*n* = 42) | | |
| MTZ | 88 (82.0) | 48 (78.7) | 34 (81) | 0.46 | 0.49 |
| CLA | 4 (3.8) | 5 (8.2) | 8 (19) | 0.004[a] | 0.09 |
| AMX | 2 (1.90) | 0 | 4 (9.5) | 0.06 | 0.02[a] |
| TET | 0 | 0 | 0 | 1 | 1 |
| Resistance ≥ 2 antibiotics % (*n*) | 3.8 (4/106) | 8.2 (5/61) | 23.8 (10/42) | 0.000[a] | 0.02[a] |

[a]*P*- values < 0.05 were significant.

TABLE 4  Clinical efficacy of triple therapy in relation to *H. pylori* susceptibility (*n* = 30)

| Antibiotics | Erradication rate (%) |
|---|---|
| Metronidazole | |
| Resistant | 21/25 (84) |
| Susceptible | 4/5 (80) |
| Rifampicin | |
| Resistant | 6/7 (85.7) |
| Susceptible | 19/23 (82.6) |
| Levofloxacin | |
| Resistant | 6/8 (75) |
| Susceptible | 19/22 (86.4) |
| Clarithromycin | |
| Resistant | 3/7 (42.8) |
| Susceptible | 22/23 (95.6) |
| Amoxicillin | |
| Resistant | 0/1 (0) |
| Susceptible | 25/29 (86.2) |
| Tetracycline | |
| Resistant | - |
| Susceptible | 25/30 (83.3%) |
| Combined resistances | |
| Metronidazole/Levofloxacin | 2/2 (100) |
| Metronidazole/Clarithromycin | 2/4 (50) |
| Metronidazole/Rifampicin | 3/3 (100) |
| Levofloxacin/Clarithromycin | 0/1 (0) |
| Metronidazole/Levofloxacin/Rifampicin | 2/3 (66.7) |
| Metronidazole/Clarithromycin/Amoxicillin | 0/1 (0) |
| Metronidazole/Clarithromycin/Levofloxacin | 1/1 (100) |

normal spiral-shaped bacillary form into the coccoid form, it allows the microorganism to survive in hostile environments and enter a viable but non-culturable (VBNC) state, minimizing its detectability by conventional culture techniques (22, 23). Transformation to coccoid form due to adverse conditions (pH, pO2, aging, aerobiosis, temperature increase, or antibiotic exposure) (24) may explain why the susceptibility profile was obtained in 75% and not 100% of *H. pylori* positive patients.

Agar dilution method is the gold standard to antibiotic resistance of *H. pylori*. However, using this method is laborious Miftahussurur et al. (25), concluded that E-test has an acceptable agreement for levofloxacin, metronidazole, tetracycline, and clarithromycin but further confirmation may be necessary for amoxicillin.

Multiple studies worldwide have confirmed the increase in therapeutic failures in patients with strains resistant to clarithromycin, finding eradication rates of only 22.2% compared to 90.2% in sensitive strains (26). Clarithromycin resistance is known to be a primary factor for treatment failure in *H. pylori* infections; hence, there have been numerous studies about tailored therapies according to a clarithromycin resistance test (27). It is recommended separating the regions, considering the prevalence of resistance to this antimicrobial and using intent-to-treat (ITT) as the first line only in those with a low prevalence of resistance (<15%) (10, 28).

The increasing *H. pylori* resistance to previously effective antibiotic treatments has become of great concern and requires careful choice of therapies and revision of therapeutic strategies.

Maastricht VI, a new focus is set on molecular testing for *H. pylori* detection and antibiotic susceptibility with support for the role of antibiotic stewardship. The most effective empirical regimens are revised if individual antibiotic resistance is not available (10).

In this study, the clarithromycin resistance rate was 19%, being 10.8% higher than that reported in the 2020 study (16), without statistically significant variation. However, this is the first report of clarithromycin resistance exceeding 15% in the west central region of Colombia after more than 10 years of resistance surveillance. It is important to avoid the use of clarithromycin-based triple regimens in countries/regions with high (>15%) primary clarithromycin resistance in *H. pylori* without susceptibility testing (10, 28). This is similar to results reported in Tumaco (20.5%) (29), Medellín (18.8%) (30), and some studies in Bogotá (13.6%–17.7%) (31). However, it is important to mention that although clarithromycin is the antibiotic associated with the most resistance studies in Colombia, the differences between the reports have been variable (31, 32). In Latin America, resistance to clarithromycin were seen in Mexico, Colombia, Argentina, and Brazil (33) and Peru (34). According to Megraud et al. (35), overall resistance to clarithromycin is above the threshold of 15% except in Indonesia where reporting is close to 9%.

According to previously published studies on *H. pylori* resistance in Colombia, the antibiotic with the highest percentage of resistance is metronidazole, which reached 72%–93% in Bogotá and around 88% in the Western Central Region of Colombia (31); however, this has been considered of not the greatest therapeutic importance (36) because it does not significantly influence eradication rates (37, 38). Resistance rate to metronidazole (81%) remained similar to the results previously published by the group (16, 17). This is like the rates reported in developing countries due to the frequent use of the drug for the treatment of common infections, mainly dental, gastrointestinal, or genital origin. According to Megraud et al. (35), overall resistance to metronidazole in adults is between 45% and 55% except in China which is 78%.

Amoxicillin resistance in Colombia has been evaluated, showing rates in the city of Bogotá in the years 2008 and 2010 of 7% and 3.8%, respectively, while in the case of Tumaco, the percentage in 2012 was 20.5% (31). In this study, the existence of resistant strains in the Western Central Region of Colombia, resistance to amoxicillin was 9.5%, which is like the values reported in the city of Bogotá, the difference found in the evaluation of the results obtained from the previous study being statistically significant (*P* = 0.02). However, Martínez et al. (33) reported that no resistant strains have been found in nine studies in Latin America, making it difficult to figure out a possible increase in resistance to amoxicillin in the region.

Levofloxacin resistance in this study was 26.2%, which is like the values reported in Colombia in the studies in the city of Bogotá (27.3%) in 2014 (37) while, as reported by Megraud et al. (35), primary overall resistance to this antibiotic in adults ranges between 14.0% and 20.0% with a trend of increase over time and variability in regions. In contrast, no isolates with tetracycline resistant, which is similar to previous results (2009 and 2020) (16, 17) in the Western Central Region of Colombia, or Europe (35), or in other areas of Colombia del 1.7% (39). However, there are exceptional cases where resistances of 50% have been reported (40). It should be noted that 15 of the 30 patients studied with 13C UBT follow-up and antimicrobial susceptibility presented combined resistance to two or more antibiotics, the main combination being metronidazole and clarithromycin (13.3%), which is consistent with what was stated by Megraud et al. (35).

Because of an increase in *H. pylori* antibiotic resistance, the eradication rate of the infection with empirical therapy has been decreasing. Tailored therapy, based on the antibiotic selection according to susceptibility testing results, was found to be superior to empiric treatment (OR 2.07, 95% CI 1.53–2.79) in the study of Rokkas et al. (41).

In Colombia, only two studies have reported multidrug-resistant strains that, and in the study by Arévalo et al. (38) they were found in patients with three or more failed treatments. While many factors are associated with the failure of *H. pylori* eradication, the main contributors are patient noncompliance and increased antimicrobial resistance, especially to quinolones and macrolides (42). In this study, we observed that in patients with clarithromycin resistance, the eradication rate decreased to <50% (Table 4), corroborating what Roberts et al. reported (42). Therefore, optimizing the first-line regimen based on local antibiotic resistance patterns is critical to prevent repeated

courses of treatment and the spread of secondary antibiotic resistance (10, 43). The increasing *H. pylori* resistance to previously effective antibiotic treatments has become of great concern and requires careful choice of therapies and revision of therapeutic strategies (10). Growing antibiotic resistance and previous unsuccessful treatment attempts impede eradication success and ease emergence and spread of multidrug (MDR "resistance to ≥3 antibiotics of different class") resistant strains (44), and this is the first report about MDR in the region.

## Conclusions

The high rate of resistance to metronidazole, clarithromycin, and levofloxacin found in this study proves the risk of therapeutic failure when using these antimicrobials in the treatment of *H. pylori* in the population from Western Central Region of Colombia. The ideal success rate in eradicating *H. pylori* infection (≥90%) was not achieved in this study. *H. pylori* MDR is the biggest challenge in the management, and this is the first time that MDR is reported in the region. Moreover, the updated percentages of resistance to clarithromycin in this geographical area have increased.

The main results of this study emphasize that patients should be treated according to local rates of antimicrobial resistance, encouraging *H. pylori* susceptibility testing not only in these cities but throughout the country. Future studies with a larger number of patients are desirable to confirm these results. The results imply that inadequate eradication therapy not only increases healthcare costs due to the need for more studies or treatments but also has a negative impact on the quantity and quality of life of affected patients.

## ACKNOWLEDGMENTS

This research was funded by Universidad Libre (Convocatoria interna 07) and Universidad Tecnológica de Pereira (Cod: CIE 5-17-4).

Conceptualization, A.A. and J.M.; methodology, A.A., Y.G., J.M., J.S., L.C., T.M, P.F., L.L., R.P. and B.A; software, P.F and R.P.; validation, A.A., T.M., B.A. and J.M.; formal analysis, A.A and P.F; investigation, A.A., J.M., B.A., and J.S.; data curation, A.A., T.M. and P.F.; resources, A.A., T.M. and P.F.; writing—original draft preparation, A.A and P.F.; writing—review and editing, A.A., Y.G., J.M., J.S., L.C., T.M, P.F., L.L., R.P. and B.A.; visualization, A.A., L.L.,T.M., B.A. and J.M.; supervision, A.A., T.M., Y.G. and Y.G.; project administration, A.A. and J.M; funding acquisition, A.A., J.S and J.M. All authors have read and agreed to the published version of the manuscript.

Informed consent was obtained from all subjects involved in the study. Written informed consent has been obtained from the patient[s] to publish this paper.

## AUTHOR AFFILIATIONS

[1]Grupo de Investigación en Microbiología y Biotecnología (MICROBIOTEC), Universidad Libre Seccional Pereira, Pereira, Colombia
[2]Grupo de Investigación ESCULAPIO, Universidad Libre Seccional Cali, Cali, Colombia
[3]Grupo de Investigación en Gerencia del Cuidado, Universidad Libre Seccional Pereira, Pereira, Colombia
[4]Grupo de Investigación en Enfermedades Infecciosas (GRIENI), Universidad Tecnológica de Pereira, Pereira, Colombia
[5]Grupo de Investigación Médica, Universidad de Manizales, Manizales, Colombia
[6]SES Hospital Universitario de Caldas, Manizales, Colombia
[7]Centro de Especialistas de Risaralda, Pereira, Colombia

## AUTHOR ORCIDs

Adalucy Alvarez-Aldana  http://orcid.org/0000-0001-7460-4163
Lina María Londoño-Giraldo  http://orcid.org/0000-0002-7824-0245

## FUNDING

| Funder | Grant(s) | Author(s) |
|--------|----------|-----------|
| Universidad Tecnológica de Pereira (UTP) | CIE 5-17-4 | Adalucy Alvarez-Aldana |
| | | Paula Andrea Fernandez Uribe |
| | | Tatiana Mejía Valencia |
| | | Yina Marcela Guaca-Gonzalez |
| | | Jorge Javier Santacruz-Ibarra |
| | | Brenda Lucia Arturo-Arias |
| | | Luis Javier Castañeda-Chavez |
| | | Robinson Pacheco-López |
| | | Lina María Londoño-Giraldo |
| | | José Ignacio Moncayo-Ortiz |

## DATA AVAILABILITY

The data presented in this study are available on request from the corresponding author of the project A.A.

## ETHICS APPROVAL

The study was conducted in accordance with the Declaration of Helsinki and approved by the Ethics Committee of Universidad Tecnológica de Pereira (protocol code CBE-SYR-162016, 12 September 2016) for studies involving humans.

## ADDITIONAL FILES

The following material is available online.

### Open Peer Review

**PEER REVIEW HISTORY (review-history.pdf).** An accounting of the reviewer comments and feedback.

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
