## [Reviewer comments · Microbiology Spectrum]

Microbiology Spectrum

Antimicrobial susceptibility of clinical *Helicobacter pylori* isolates and its eradication by standard triple therapy: A study in west central region of Colombia

Adalucy Alvarez Aldana, Paula Fernandez-Uribe, Tatiana Mejia-Valencia, Yina Guaca-Gonzalez, Jorge Javier Santacruz-Ibarra, Brenda Arturo-Arias, Luis Castañeda-Chavez, Robinson Pacheco-López, Lina Londoño-Giraldo, and José Ignacio Moncayo Ortiz

Corresponding Author(s): Adalucy Alvarez Aldana, Universidad Libre - Campus Pereira

Review Timeline:

Submission Date:	February 19, 2024
Editorial Decision:	April 8, 2024
Revision Received:	April 25, 2024
Accepted:	April 26, 2024

Editor: Eleanor Powell

Reviewer(s): Disclosure of reviewer identity is with reference to reviewer comments included in decision letter(s). The following individuals involved in review of your submission have agreed to reveal their identity: Vinoj Gopalakrishnan (Reviewer #1)

Transaction Report:

DOI: <https://doi.org/10.1128/spectrum.00401-24>

Re: Spectrum00401-24 (Antimicrobial susceptibility of clinical *Helicobacter pylori* isolates and its eradication by standard triple therapy: A study in west central region of Colombia)

Dear Prof. Adalucy Alvarez Aldana:

Thank you for the privilege of reviewing your work. After reviewing feedback from reviewers, modifications are required before potential publication. Below you will find my comments, instructions from the Spectrum editorial office, and the reviewer comments.

Revision Guidelines

Sincerely,
Eleanor Powell
Editor
Microbiology Spectrum

Reviewer #1 (Comments for the Author):

This is an interesting manuscript "Antimicrobial susceptibility of clinical *Helicobacter pylori* isolates and its eradication by standard triple therapy: A study in west central region of Colombia". Given the rising incidence of antibiotic resistance in *H. pylori*, consider research to be crucial for the population.

Overall, the experiments are performed well, and manuscript was well written, I recommend this research manuscript is appropriate to publish with Major revision mentioned below.

1. The author would think about in addition to Gram staining and biochemical testing, the polymerase chain reaction (PCR) is an extremely sensitive and specific technique for diagnosing *Helicobacter pylori*. So, the study will be more reliable.

2. On what basis *Helicobacter pylori* strains were chosen?

Reviewer #2 (Comments for the Author):

This is a generally well-written article describing a small study on Triple therapy of *Helicobacter pylori* in Colombia. It presents results of a PPI-clarithromycin-amoxicillin regimen for treatment of patients with gastritis and includes data on efficacy, eradication and antibiotic susceptibility of drugs used for *H. pylori*. The main take home is that clarithromycin resistance is growing and susceptibility to clarithromycin is strongly correlated with eradication. It is recommended that empirical treatment of *H. pylori* be based on knowledge of local antibiotic resistance patterns. The methods used are appropriate and well described. There are a few instances of rather informal English which should be corrected:

Line 64 "figured out as" should be changed to identified as

Line 74 "apparition" should be changed to appearance or emergence

Line 103 "figure out" should be changed to discover

Dear, Editors:

Microbiology Spectrum - ASM

Please find the revised version of our paper "**Antimicrobial susceptibility of clinical *Helicobacter pylori* isolates and its eradication by standard triple therapy: A study in west central region of Colombia**" co-authored by Adalucy Alvarez-Aldana *et al.*, which we would like to submit for publication in your Journal. It contains original results which have not been submitted for publication elsewhere.

Response to Reviewers,
Microbiology Spectrum
Manuscript ID: Spectrum00401-24R1

Thank you for your interest on our paper. As recommended, please find hereafter the list of the whole changes amended in our manuscript "**Antimicrobial susceptibility of clinical *Helicobacter pylori* isolates and its eradication by standard triple therapy: A study in west central region of Colombia**" co-authored by Adalucy Alvarez-Aldana et al (Manuscript ID: Spectrum00401-24R1) according to your comments for the format of the manuscript. You will find all modifications done in new version of "Marked-Up Manuscript" using **blue highlight** and will uploaded a clean .DOC/.DOCX version of the revised manuscript and removed the previous version.

Reviewer #1:

This is an interesting manuscript "Antimicrobial susceptibility of clinical *Helicobacter pylori* isolates and its eradication by standard triple therapy: A study in west central region of Colombia". Given the rising incidence of antibiotic resistance in *H. pylori*, consider research to be crucial for the population. Overall, the experiments are performed well, and manuscript was well written, I recommend this research manuscript is appropriate to publish with Major revision mentioned below.

Comment 1: The author would think about in addition to Gram staining and biochemical testing, the polymerase chain reaction (PCR) is an extremely sensitive and specific technique for diagnosing *Helicobacter pylori*. So, the study will be more reliable.

Answer 1: Despite the availability of multiple diagnostic methods, selection of one or more diagnostic tests will depend on the several conditions (e.g sensitivity and specificity). Although PCR method is beneficial to detect *H. pylori*, the following points were taken into account:

-Cultures from gastric biopsies are the gold standard (Bordin et al, 2021)¹.

- Culture-based diagnostics followed by phenotypic antimicrobial susceptibility testing is the gold standard for detection of resistance patterns (Shakir et al, 2023)².

Since the objective of the research was to follow up a cohort with respect to the eradication rate using triple standard therapy, it was necessary to obtain the resistance patterns of the most reliable method (gold standard) to determine if the treatment failure was due to bacterial resistance. The results of biochemical tests and Gram staining of presumptive *H. pylori* colonies did not generate doubts, therefore, molecular confirmation (PCR) was not performed.

Comment 2: On what basis *Helicobacter pylori* strains were chosen?

Answer 2: As described in the subtitles of Materials and Methods, “Patient Population”, “*H. pylori* Culture” and “E-Test and Definition of Susceptibility Testing”, all *H. pylori* isolates obtained by the primary culture were subcultured on non-selective Tryptic soy agar and the isolates subcultured were profiled for antimicrobial resistance by E-test. Therefore, the selection of the strains was based on the positive growth in culture.

Reviewer #2:

This is a generally well-written article describing a small study on Triple therapy of *Helicobacter pylori* in Colombia. It presents results of a PPI-clarithromycin-amoxicillin regimen for treatment of patients with gastritis and includes data on efficacy, eradication and antibiotic susceptibility of drugs used for *H. pylori*. The main take home is that clarithromycin resistance is growing and susceptibility to clarithromycin is strongly correlated with eradication. It is recommended that empirical treatment of *H. pylori* be based on knowledge of local antibiotic resistance patterns. The methods used are appropriate and well described.

Comment 1: There are a few instances of rather informal English which should be corrected:
Line 64 "figured out as" should be changed to identified as
Line 74 "apparition" should be changed to appearance or emergence
Line 103 "figure out" should be changed to discover

Answer 1: All sentences were corrected in the manuscript.

¹ Bordin, D. S., Voynovan, I. N., Andreev, D. N., & Maev, I. V. (2021). Current Helicobacter pylori Diagnostics. Diagnostics (Basel, Switzerland), 11(8), 1458. <https://doi.org/10.3390/diagnostics11081458>

² Shakir, S. M., Shakir, F. A., & Couturier, M. R. (2023). Updates to the Diagnosis and Clinical Management of Helicobacter pylori Infections. Clinical chemistry, 69(8), 869–880. <https://doi.org/10.1093/clinchem/hvad081>

We thank you in advance for the consideration given this paper. We believe that with the suggestions made, the manuscript has improved considerably. We look forward to a favourable response from you.

Adalucy Álvarez – Aldana. BSc, MSc, Ph.D.
Universidad Libre – Colombia
Corresponding Autor

Re: Spectrum00401-24R1 (Antimicrobial susceptibility of clinical *Helicobacter pylori* isolates and its eradication by standard triple therapy: A study in west central region of Colombia)

Dear Prof. Adalucy Alvarez Aldana:

It is my pleasure to inform you that your manuscript has been accepted, and I am forwarding it to the ASM production staff for publication. Your paper will first be checked to make sure all elements meet the technical requirements. ASM staff will contact you if anything needs to be revised before copyediting and production can begin. Otherwise, you will be notified when your proofs are ready to be viewed.

Sincerely,
Eleanor Powell
Editor
Microbiology Spectrum